# COMPOUND TOKENS: CHANNEL FUSION FOR VISION-LANGUAGE REPRESENTATION LEARNING

**Maxwell Mbabilla Aladago** [*]
Department of Computer Science
Dartmouth College, Hanover, NH 03755
{maxwell.m.aladago.gr}@dartmouth.edu

**AJ Piergiovanni**
Google Research

## ABSTRACT

We present an effective method for fusing visual-and-language representations for several question answering tasks including visual question answering and visual entailment. In contrast to prior works that concatenate unimodal representations or use only cross-attention, we compose multimodal representations via channel fusion. By fusing on the channels, the model is able to more effectively align the tokens compared to standard methods. These multimodal representations, which we call compound tokens are generated with cross-attention transformer layers. We demonstrate the effectiveness of compound tokens using an encoder-decoder vision-language model trained end-to-end in the open-vocabulary setting. Compound Tokens achieve highly competitive performance across a range of question answering tasks including GQA, VQA2.0, and SNLI-VE.

## 1 INTRODUCTION

The two most common strategies for fusion multimodal representations are called merged attention and co-attention, illustrated in Figures 2a and 2b respectively. Merged attention simply concatenates the different unimodal representations *along the sequence dimension* (Luowei et al., 2019; Hendricks et al., 2021; Dou et al., 2022), while co-attention processes unimodal tokens in different transformer encoders where cross-attention is used to exchange information between the two modalities (Bugliarello et al., 2021; Li et al., 2021; Dou et al., 2022). These two methods have inbuilt limitations: merged attention does not have the benefit of cross-attention for token alignment while co-attention does not enjoy a global receptive field across all vision and text tokens.

We introduce Compound Tokens that address these limitations in an efficient and simple pipeline. Compound Tokens use the tokens from one modality to query the other modality, and concatenate the output with the query tokens *along the channels*. Channel concatenation (illustrated in Figure 2c) does not increase the token length which makes the method efficient. This approach is also more effective than summation and element-wise product as shown in Table 2. Our method creates vision-to-text compound tokens and text-to-vision compound tokens. These representations are then contacted along the token dimension for further modeling. Compound Tokens outperform both merged attention and co-attention on GQA (Hudson & Manning, 2019), SNLI-VE (Xie et al., 2019), and VQA (Goyal et al., 2017) as shown in Table 1. Our method also beat METER (Dou et al., 2022) by $2.26\%$ on SNLI-VE and CFR (Nguyen et al., 2022) by $8.83\%$ on GQA. Table 3 shows compound tokens to be competitive among existing state-of-the-art models.

## 2 COMPOUND TOKENS

We now introduce our multimodal fusion method more formally. Our method, illustrated in Figures 2c & 1, draws from both co-attention and merged-attention. Given a sequence of $N$ image tokens, $\mathcal{I} \in \mathbb{R}^{N \times d}$, and $M$ text tokens, $\mathcal{T} \in \mathbb{R}^{M \times d}$, Compound Tokens first projects the visual and language tokens into half of the embedding space so that the total number of features is maintained after channel concatenation: $\widetilde{\mathcal{I}} \in \mathbb{R}^{N \times \frac{d}{2}}$; $\widetilde{\mathcal{T}} \in \mathbb{R}^{M \times \frac{d}{2}}$ for the image and text tokens respectively.

---

[*]Work done while interning at Google.

Next, we employ only two cross-attention layers (unlike co-attention (Dou et al., 2022) that uses cross-attention and self-attention in every block) to create visual and language compound tokens

$$\widehat{\mathcal{I}} = \mathcal{A}\left(\widetilde{\mathcal{I}}, \widetilde{\mathcal{T}}, \widetilde{\mathcal{T}}\right) \qquad \in \mathbb{R}^{N \times \frac{d}{2}} \qquad \widehat{\mathcal{T}} = \mathcal{A}\left(\widetilde{\mathcal{T}}, \widetilde{\mathcal{I}}, \widetilde{\mathcal{I}}\right) \qquad \in \mathbb{R}^{M \times \frac{d}{2}} \qquad (1)$$

$$\mathcal{I}_{cmpd} = \text{C-Concat}\left(\widetilde{\mathcal{I}}, \widehat{\mathcal{I}}\right) \qquad \in \mathbb{R}^{N \times d} \qquad \mathcal{T}_{cmpd} = \text{C-Concat}\left(\widetilde{\mathcal{T}}, \widehat{\mathcal{T}}\right) \qquad \in \mathbb{R}^{M \times d}, \qquad (2)$$

where $\mathcal{A}(q, k, v)$ is the cross-attention function with $q$, $k$, and $v$ as queries, keys, and values respectively. C-Concat$(u, v)$ concatenates tensors $u$ and $v$ along the feature dimension. We combine vision-to-text compound tokens $\mathcal{I}_{cmpd}$, and text-to-vision compound tokens $\mathcal{T}_{cmpd}$, into a set of output compound tokens as in merged attention architectures

$$\mathcal{O}_{cmpd} = \text{Concat}\left(\mathcal{I}_{cmpd}, \mathcal{T}_{cmpd}\right) \in \mathbb{R}^{(N+M) \times d}. \qquad (3)$$

Following previous methods, we feed $\mathcal{O}_{cmpd}$ into a multimodal encoder before generating the outputs with an auto-regressive decoder.

## 3  MAIN RESULTS

In control experiments with and without vision-language pretraining shown in Table 1, our method outperforms merged attention and co-attention on SNLI-VE, and GQA. Experimental details are provided in Section C.

Table 1: **Comparisons with other fusion methods**: Compound Tokens outperform merged attention and co-attention Dou et al. (2022) with, and without vision-language pretraining (VLP). Params shows the number of parameters in the entire model; $L$ is the total number of layers in the fusion encoder.

| Fusion Method | $L$ | Params ($\times 10^6$) | *With* VLP | | *Without* VLP | |
|---|---|---|---|---|---|---|
| | | | SNLI-VE | GQA | SNLI-VE | GQA |
| Merged Attention | 12 | 332.94 | 81.78 | 78.13 | 79.81 | 78.07 |
| Co-Attention | 12 | 361.26 | 80.50 | 75.92 | 80.20 | 77.75 |
| Compound Tokens (Ours) | 12 | 339.97 | **82.47** | **79.55** | **80.52** | **78.21** |

## 4  CONCLUSION

We introduce Compound Tokens, a new multimodal fusion method for vision-and-language representation learning that is on par with several competitive multimodal models on multiple visual question answering tasks (See Table 3). We hope that this new perspective of merging representations from different modalities will spur research for more effective and efficient fusion approaches in multimodal settings.

### URM STATEMENT

Author Maxwell Mbabilla Aladago meets the URM criteria of ICLR 2023 Tiny Papers Track.

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

APPENDIX

## A ARCHITECTURE

We use ResNet-50 (He et al., 2015) as our image encoder and T5-base (Raffel et al., 2020) as our text encoder in an encoder-decoder architecture. The outputs of the image and text encoders are provided to our novel fusion method described in Section 2. A T5-base decoder consumes the output of the fusion module and generates free form text for all question answering tasks. The image encoder is pretrained on ImageNet-1k (Deng et al., 2009) while the text encoder and decoder use pretrained T5 weights.

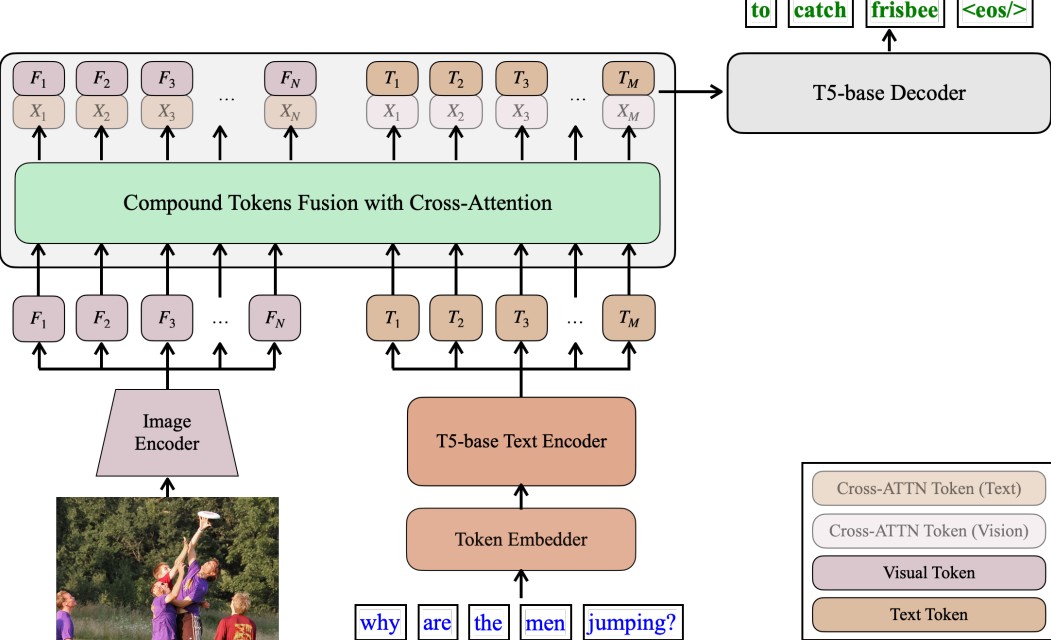

Figure 1: **Model Architecture:** Compound Tokens Fusion is illustrated in Figure 2c. ResNet-50 (He et al., 2015) and T5-base (Raffel et al., 2020) are used for the image and text encoders respectively.

### A.1 ILLUSTRATION OF COMPOUND TOKENS

In this section, we provide an illustration of our fusion method, along with other popular fusion methods namely merged attention and co-attention.

## B WHY CHANNEL CONCATENATION

To determine the best way of composing compound tokens, we examined a number of options with a prime objective to not increase the token length. To this end, we sampled four combination methods and compared them on SNLI-VE and GQA as the performances on these datasets in our setup are more stable compared to VQA. Given input queries $q$ and cross-attention layer's outputs $X$, we explored the following: (1) *channel concatenation* where we concatenate $q$ and $X$ along the feature dimension as described in Section 2. (2) *weighting* uses the operation $Y = \alpha q + \beta X$ where $\alpha$ and $\beta$ are learnable scalars initialized randomly, and $Y$ is the output. (3) In *Element-wise product*, $Y = q \odot X$. (4) Finally, we tested a simple summation of the tensors, $Y = q + X$. All these methods use approximately the same number of flops and parameters. The results in Table 2 show channel concatenation is better than the other methods, hence our use of channel concatenation in the rest of the paper.

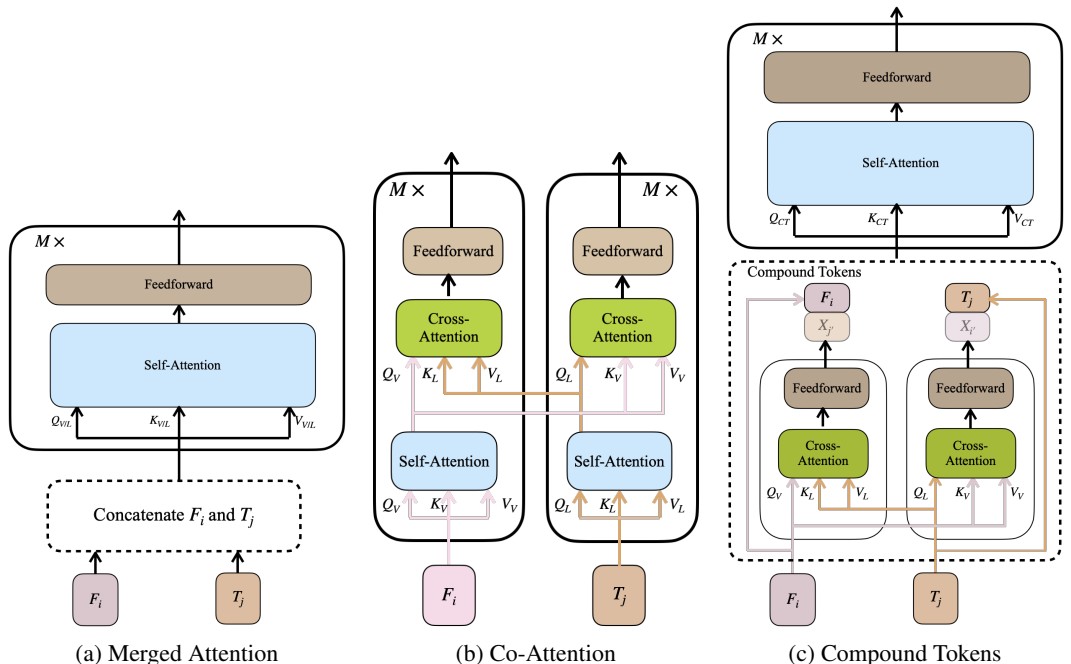

(a) Merged Attention         (b) Co-Attention         (c) Compound Tokens

Figure 2: **Different multimodal fusion methods**: Illustrations of two types of fusions methods in previous works: (a) co-attention, and (b) merged attention from the perspective of one visual token $F_i$, and one text token $T_j$. Our proposed compound tokens fusion method is illustrated in (c). Note that we use only one cross-attention layer for each modality compared to co-attention which uses both cross-attention and self-attention in all blocks. We concatenate the input query to the cross-attention module with the cross-attention output along the channel dimension. $Q$, $K$, and $V$ denote the input query, keys and values respectively to the attention module. $X$ represents the cross-attention layer's output. Finally, the subscripts $V$, $L$, and $CT$ respectively identify an input as visual features, text features or compound tokens, e.g., $Q_V$ indicates an input query that is composed of visual tokens.

Table 2: **Different Methods of Formulating Compound Tokens**: Channel concatenation obtains the highest accuracy on SNLI-VE and GQA.

| Method | GFlops | SNLI-VE | GQA |
|---|---|---|---|
| Channel Concatenation | 20.71 | **80.85** | **80.79** |
| Weighting | 20.71 | 80.63 | 80.61 |
| Summation | 20.71 | 80.75 | 80.35 |
| Element-wise Product | 20.71 | 80.81 | 78.31 |

### B.1 COMPARISON WITH EXISTING APPROACHES

We compare our model with various competitive recent models such as METER (Dou et al., 2022), ALBEF (Li et al., 2021), and CFR (Nguyen et al., 2022). The models in Table 3 generally have approximately the same number of parameters, but differ significantly on the pretraining datasets, pretraining objectives, and backbone encoders. For example, while we use Conceptual Captions (Sharma et al., 2018) and COCO (Lin et al., 2014) as our pretraining datasets, METER used Conceptual Captions, COCO, Visual Genome (Krishna et al., 2016) and SBU Captions (Ordonez et al., 2011). ALBEF used all the datasets in METER in addition to Conceptual Captions 12M (Changpinyo et al., 2021).

The model we use for this comparison has 340 million parameters in total. We pretrain it for 500k iterations with a batch-size of 512 using an image resolution of $224 \times 224$ and further finetune for

200k iterations on each of the downstream tasks at resolution $384 \times 384$ with batch size 128. This model uses a multimodal encoder with 12 blocks.

Except for SimVLM (Wang et al., 2022) which has about 1.5 billion parameters and uses a significantly large pretraining data (a 1.8 billion private dataset), our model outperforms all other methods on SNLI-VE and GQA by large margins. We are confident that further pretraining and increasing image resolution will improve our already competitive result on the VQA dataset. Scaling up the model may also yield additional performance improvements.

Table 3: **Comparison with SOTA:** Compound Tokens outperforms all other models on SNLI-VE and GQA in an open-vocabulary evaluation except SimVLM (Wang et al., 2022) which used a private dataset of 1.5B samples. For VQA, we present the results in the closed-vocabulary setting for fair comparisons with the other methods: our open-set evaluation is significantly worse than the closed-set evaluation model on this task. The best values among the models besides SimVLM are in **bold**. The second best values are underlined. *The flops are based on our calculations. Our model is extremely more efficient than the rest partly because we use a short text sequence length of 32 and a ResNet-50 backbone that produces 49 visual tokens. Results in gray denote large models trained on substantially more data than our model.

| Approach | Params | GFlops* | VQA | SNLI-VE | GQA |
|---|---|---|---|---|---|
| SimVLM$_{Huge}$ (Wang et al., 2022) | 1.5B | 890 | *80.34* | *86.32* | - |
| VisualBERT (Li et al., 2019) | | | 66.70 | 75.69 | - |
| UNITER (Chen et al., 2020) | | | 73.82 | 79.39 | - |
| LXMERT (Tan & Bansal, 2019) | | | 69.90 | - | 60.00 |
| ALBEF (Li et al., 2021) | 418M | 122 | 75.84 | 80.91 | - |
| METER (Dou et al., 2022) | 336M | 130 | **77.68** | 80.61 | - |
| BLIP (Li et al., 2022) | 475M | 122 | 77.54 | - | - |
| 12-in-1 (Lu et al., 2020) | | | 71.30 | - | 60.50 |
| VinVL (Zhang et al., 2021) | | | 75.95 | - | 65.05 |
| VL-T5 (Cho et al., 2021) | | | 70.30 | - | 60.80 |
| CFR (Nguyen et al., 2022) | | | 69.80 | - | 73.60 |
| Compound Tokens (Ours) | 340M | 36 | 70.62 | **82.87** | **82.43** |

## C  EXPERIMENTAL DETAILS

### C.1  DATASETS

**SNLI-VE** (Xie et al., 2019) is a dataset of approximately 500,000 image-text pairs used for visual entailment (VE). Given an image and a proposed statement, the task for this dataset requires determining whether the statement is neutral, entails, or contradicts the image.

**Visual Question Answering (VQA2.0)** (Goyal et al., 2017) is a widely used benchmark for many question-answering models and contains 400,000 image-text pairs spanning 3,130 output categories. Each image-question pair is associated with 10 answers.

**GQA** (Hudson & Manning, 2019) is a vision question answering dataset of complex compositional questions comprising scene-object relations formed from Visual Genome (Krishna et al., 2016) with approximately 22 million question-answer pairs and 113 thousand images.

We emphasize that for all tasks, our model generates a correct answer in an open-vocabulary setting of about 32,000 words irrespective of the number of categories in the task. A generated response is counted as correct if and only if it matches exactly with the ground-truth answer. We use the VQA metric[1] for VQA2.0 and simple accuracy for GQA and SNLI-VE as evaluation metrics.

In addition to the downstream datasets, we also use CC3M[2] (Sharma et al., 2018) and COCO Captions (Lin et al., 2014) for pretraining. The pretraining setup uses a mixture of these datasets across

---

[1]https://visualqa.org/evaluation.html
[2]The version of the dataset we used has about 2 million samples

four objectives: (1) **image-text matching** where the model predicts whether an image-text pair is a match or not, (2) **captioning** where the model generates the full caption given an image, (3) **caption completion** where the model completes a masked caption, and (4) **masked-language modeling** as in BERT (Devlin et al., 2019).

## C.2 Hyper-parameter settings

We provide full details of our hyper-parameter settings in this section. We use Adam (Kingma & Ba, 2015) to optimize all our models. The learning rate starts from zero and warms up linearly to the base rate after 8k iterations. Cosine annealing (Loshchilov & Hutter, 2017) with a cycle rate of 100k steps is used to decay the rate to zero by the end of training. We use gradient clipping with a maximum norm of 1.0 in all our experiments.

We do not use any data augmentation beyond resizing and normalization in all the ablation experiments and finetuning experiments. We apply random cropping and AutoAugment (Cubuk et al., 2019) during pretraining of our main model.

All our pretraining experiments use a batch size of 512 and image resolution $224 \times 224$. The batch size is divided equally among the four pretraining objectives: image captioning, caption completion, image-text matching, and masked language modeling. We also sample the same number of examples from CC3M and COCO in every iteration. The batch size and resolutions are set to 128, and $384 \times 384$ respectively whenever training from scratch or finetuning.

The datasets we used and our model are described in Section C. The rest of the hyper-parameters are listed in Table 4.

Table 4: **Hyper-parameter Settings**: We enumerate the hyper-parameters for our ablation experiments and main model. $L$ is the number of blocks in a multimodal encoder. Main Model is the model we used in Table 3 for comparison with existing works.

| Experiment | Phase | $L$ | Iterations | LR | Dropout | Weight Decay |
|---|---|---|---|---|---|---|
| Ablations | Pretraining | 0 / 12 | 300k | $1.1e^{-4}$ | $1e^{-3}$ | 0.1 |
| | Finetuning | 0 / 12 | 100k | $5e-5$ / $3.1e^{-3}$ | 0 / $1e^{-3}$ | $1e^{-4}$ |
| | Scratch | 0 / 12 | 300k | $7.5e^{-5}$ / $3e^{-5}$ | $1e^{-2}$ | $1e^{-3}$ |
| Main Model | Pretraining | 12 | 500k | $1.1e^{-4}$ | $1e^{-3}$ | 0.1 |
| | Finetuning | | 200k | $3e^{-5}$ | | $1e^{-4}$ |

