# OpenReview forum: "Compound Tokens: Channel Fusion for Vision-Language Representation Learning"
_ICLR.cc/2023/TinyPapers — Submitted to Tiny Papers @ ICLR 2023_

### Official Review · Reviewer_1yNm · 2023-03-20

**Confidence:** 4

**Summary Of Contributions:**

This paper proposes a method for fusing visual-and-language (VL) representations to learn various VL tasks. Instead of simple concatenation of unimodal representations or using cross-attention, it composes multimodal representations via channel fusion, which is referred to as “compound tokens”. This paper shows the effectiveness of compound tokens through experiments using an encoder-decoder VL model on GQA, VQA2.0, and SNLI-VE — both with and without VL pre-training (VLP).

**Rating:**

High Impact (HI): a submission which meets the reviewing criteria and is predicted to make an impact on the field

**Strengths And Weaknesses:**

Strengths
- The paper is well-written and easy to read.
- The paper clearly explains the methodology of compound tokens and provides empirical comparisons with other fusion methods.
- Experimental settings and results are described in detail and thoroughly.
- Justification of methodology choice is supported by experimental results.

Weaknesses
- The content of the paper is too long to fit in ICLR Tiny Papers’ 2-page limit. Some of the main content is moved to the appendix, including the figure of the architecture, the figure of comparison with other VL fusion methods, the result table of comparison with SOTA, etc. Moving these details to the appendix oversimplifies the extent of the analysis conducted.
- What kind of problem motivates this paper / this paper wants to tackle is a bit unclear.

**Suggested Changes:**

- Despite compound tokens being the proposed method, it hasn’t been evaluated using other image encoders and text encoders aside from ResNet-50 and T5-base. It would be interesting to see if compound tokens consistently outperform other fusion methods across different image encoders and text encoders.
- Is there a missing statement to explain the grey italicized writing in Table 3?

---

### Meta-Review · Area_Chair_z2KG · 2023-04-05

**Recommendation:** Invite to present
**Confidence:** 4

**Metareview:**

This paper introduces a new multimodal fusion method called Compound Tokens for vision-and-language representation learning.

Experimental results show that the proposed method is better than existing merged attention and co-attention approaches.
There is only one reviewer for this paper. To assess the quality, clarity, originality, and significance of this work, the AC has also read the paper and merged the opionions to make the recommendation.

This paper generally meets the CCR review standard. The paper is well-written, and the experimental results support the claims. Some important details are placed in the Appendix; it is suggested to move them (e.g., a simplified figure of the architecture and result discussion with SOTA methods). It is possible to make the abstract and introduction more concise to fit the page limit.

**Summary:**

The paper can be further enhanced with additional analysis using other image encoders and text encoders aside from ResNet-50 and T5-base.

**Reason For Not Giving A Higher Recommendation:**

N/A

**Reason For Not Giving A Lower Recommendation:**

The paper is generally sound with necessary details.

---

### Decision · Program_Chairs · 2023-04-10

Invite to present (notable)